# Human PARP1 Facilitates Transcription through a Nucleosome and Histone Displacement by Pol II In Vitro

**DOI:** 10.3390/ijms23137107

**Published:** 2022-06-26

**Authors:** Elena Y. Kotova, Fu-Kai Hsieh, Han-Wen Chang, Natalia V. Maluchenko, Marie-France Langelier, John M. Pascal, Donal S. Luse, Alexey V. Feofanov, Vasily M. Studitsky

**Affiliations:** 1Fox Chase Cancer Center, Philadelphia, PA 19111, USA; elena.kotova@fccc.edu (E.Y.K.); hanwench22@gmail.com (H.-W.C.); 2Department of Molecular Biology, Massachusetts General Hospital, Boston, MA 02114, USA; fukai.hsieh@gmail.com; 3Department of Biochemistry and Molecular Pharmacology, New York University Grossman School of Medicine, New York, NY 10016, USA; 4Howard Hughes Medical Institute, New York University Grossman School of Medicine, New York, NY 10016, USA; 5Biology Faculty, Lomonosov Moscow State University, 119234 Moscow, Russia; maluchenko@gmail.com (N.V.M.); avfeofanov94@gmail.com (A.V.F.); 6Department of Biochemistry and Molecular Medicine, Université de Montréal, Montréal, QC H3T 1J4, Canada; mf.langelier@umontreal.ca (M.-F.L.); john.pascal@umontreal.ca (J.M.P.); 7Department of Cardiovascular and Metabolic Sciences, Lerner Research Institute, Cleveland Clinic, Cleveland, OH 44195, USA; lused@ccf.org; 8Shemyakin-Ovchinnikov Institute of Bioorganic Chemistry, Russian Academy of Sciences, 117997 Moscow, Russia

**Keywords:** poly(ADP)-ribose polymerase-1, PARP1, nucleosome, transcription, elongation, olaparib

## Abstract

Human poly(ADP)-ribose polymerase-1 (PARP1) is a global regulator of various cellular processes, from DNA repair to gene expression. The underlying mechanism of PARP1 action during transcription remains unclear. Herein, we have studied the role of human PARP1 during transcription through nucleosomes by RNA polymerase II (Pol II) in vitro. PARP1 strongly facilitates transcription through mononucleosomes by Pol II and displacement of core histones in the presence of NAD+ during transcription, and its NAD+-dependent catalytic activity is essential for this process. Kinetic analysis suggests that PARP1 facilitates formation of “open” complexes containing nucleosomal DNA partially uncoiled from the octamer and allowing Pol II progression along nucleosomal DNA. Anti-cancer drug and PARP1 catalytic inhibitor olaparib strongly represses PARP1-dependent transcription. The data suggest that the negative charge on protein(s) poly(ADP)-ribosylated by PARP1 interact with positively charged DNA-binding surfaces of histones transiently exposed during transcription, facilitating transcription through chromatin and transcription-dependent histone displacement/exchange.

## 1. Introduction

Poly(ADP-ribosyl)ation (PARylation) is an essential covalent protein modification introduced by various poly-ADP-ribose polymerases (PARPs), a family of enzymes that transfer adenosine diphosphate ribose (ADP-ribose) from NAD+ onto a variety of proteins [1,2]. PARP1 is an abundant multi-domain protein, localized in cell nuclei that is responsible for at least 85% of cellular PARP activity [3]. It plays a role in a variety of cellular processes, including DNA repair, chromatin organization and transcription (see [1,4,5] for review). PARP1 binding to damaged DNA induces a conformational change in the protein [6,7,8], which results in its DNA-dependent activation and PARylation of target proteins (including automodification of PARP1 [7]). Core histones [9] and linker histone H1 [10] are among the targets for PARylation. Some inhibitors of PARP1 enzymatic activity are important anticancer compounds [11,12]. In particular, inhibition of PARP1 by olaparib induces synthetic lethality in tumor cells deficient in enzymes involved in homologous recombination DNA repair [13,14]. 

The members of the PARP family of enzymes participate in regulation of gene expression [15,16,17,18,19,20,21], see [1,15] for review. In particular, PARP1 is required for H1 displacement and gene de-repression, as a prerequisite for further nucleosome remodeling at promoters [22]. PARP1 also promotes gene expression by increasing the activity of transcription factors and by affecting the proteins involved in DNA methylation [23,24] and participates in release of the promoter-proximal pausing through ADP-ribosylation of NELF-E [20]. In Drosophila, PARP1 also participates in both transcription-dependent and transcription-independent histone removal from the body of actively transcribed genes [18,25,26,27]. The role of human PARP1 in histone removal and exchange during transcription by RNA polymerase II (Pol II) has not been addressed in vitro.

Herein, we utilized in vitro approaches for analysis of the effect of human PARP1 and PARylation on the efficiency of transcription through chromatin and the accompanying histone displacement/exchange. Our data suggest that both PARP1 protein and its catalytic activity are required to facilitate transcription through chromatin and transcription-dependent histone displacement during transcription. Accordingly, anti-cancer drug and catalytic inhibitor of PARP1 olaparib strongly represses PARP1-dependent transcription in vitro.

## 2. Results

### 2.1. Experimental Approach for Analysis of the Effect of PARP1 on Transcription through a Nucleosome

To study the function of human PARP1 during transcript elongation, the well-established in vitro transcription assay system, described previously, is employed [28,29] (Figure 1a). This experimental system faithfully recapitulates many key characteristics of the mechanism of transcription through chromatin in vivo [30,31,32]. In brief, yeast RNA polymerase II (yPol II) elongation complexes (ECs) were assembled using synthetic DNA and RNA oligonucleotides, and ligated to nucleosomes reconstituted on the 603 positioning sequences (Appendix A) that were extensively characterized and used previously for analysis of the mechanism of transcription through nucleosomes [28,29,33,34,35]. The ECs were incubated in the presence of a limited subset of NTPs and [α-^32^P] GTP (-UTP reaction) to advance yPol II to the position -80 (EC-80) along the template (the number indicates the distance of the active center of the enzyme from the promoter-proximal nucleosome boundary) and to pulse-label RNA. Then, the templates were incubated in the absence or presence of PARP1 and NAD+, and transcribed in the presence of an excess of all unlabeled NTPs (Figure 1a).

PARP1 is the enzyme responsible for poly(ADP-ribosy)lation (PARylation) of the target proteins; PARP1 itself is one of the primary targets for auto PARylation. Consistently, in our transcription reaction PARP1 is heavily self-PARylated in NAD+-dependent manner (Appendix A). The other PARP1 target proteins present during transcription in vitro are core histones [9,37] and yPol II [37]. Indeed, we have detected modification of at least some of core histones in the presence of biotinylated NAD+ (Appendix A); it was more difficult to evaluate the level of yPol II PARylation because potentially PARylatable Pol II subunits co-migrate with PARylated PARP1 (Appendix A and [37]).

### 2.2. PARP1 Facilitates Transcription through the Nucleosome by Pol II

As expected, during transcription of the nucleosomal templates in the absence of PARP1, characteristic pauses were observed at the positions +(11–15), +(26–27), +(35–37) and +(45–48) bp from the promoter-proximal nucleosome boundary (Figure 1b) [28]. Transcription in the presence of non-PARylated PARP1 (in the absence of NAD+) results in a moderate decrease in the overall efficiency of nucleosome traversal by yPol II and an increase of pausing at positions +(28–29) and +(38–40) (Figure 1b). The data suggest that binding of intact PARP1 to the nucleosome results in formation of the PARP1-nucleosome complex [38,39,40] that moderately inhibits progression of yPol II through the nucleosome.

In contrast, in the presence of PARP1/NAD+, when PARP1 was able to PARylate itself (Appendix A), core histones (Appendix A) and possibly yPol II subunits [37], all four major pauses were partially relieved, with an accompanying considerable increase in the yield of the run-off transcript (Figure 1b). The data suggest that PARP1-induced PARylation strongly facilitates yPol II transcription through the nucleosome by relieving all barriers along the nucleosome. 

The experiments described above were conducted using yeast yPol II; yPol II is an enzyme that shares many important properties relevant for transcription through chromatin with human Pol II (hPol II) [29]. However, yeast do not have PARP enzymes, raising the question whether yPol II is a suitable model for analysis of PARP1 function in transcription. Accordingly, our model system was adopted to allow transcription by hPol II using a very similar protocol involving assembly of hPol II ECs on synthetic DNA and RNA oligonucleotides, and ligation of the EC to 603 nucleosomes (see Methods). 

Previously, transcription through a nucleosome was conducted using promoter-initiated hPol II [29]. In comparison with the promoter-initiated transcription by hPol II and with transcription by assembled yPol II EC (Figure 1), transcription through the 603 nucleosome by assembled hPol II EC occurs less efficiently (Appendix A). However, the nucleosomal pausing pattern that is typical for the 603 nucleosome was detected, indicating that the assembled hPol II EC utilizes a similar mechanism of transcription through the nucleosome as yPol II EC (Appendix A). The differences in the efficiency of transcription by promoter-initiated and assembled hPol II ECs is likely explained by phosphorylation of hPol II that occurs during initiation from the AdML promoter used in the previous studies [29]. In the presence of PARP1/NAD+, hPol II encounters a lower nucleosomal barrier and more efficiently traverses through the nucleosome (Figure 2a), as was observed with yPol II (Figure 2b). The data show that PARP1/NAD+ similarly affects transcription through the nucleosome by hPol II and yPol II.

### 2.3. Catalytic Activity of PARP1 Is Required to Facilitate Transcription through the Nucleosome

To further evaluate the possibility that PARP1-induced PARylation strongly facilitates yPol II transcription through the nucleosome, we used a mutant PARP1 enzyme with the catalytic activity compromised by the mutation E988K [41]. Transcription in the presence of the mutant PARP1/NAD+ results in a moderate decrease in the overall efficiency of nucleosome traversal by yPol II and increase in nucleosomal pausing (Figure 2b). Thus, PARP1 E988K/NAD+ and WT PARP1 without NAD+ similarly (and very differently from WT PARP1/NAD+) affect transcription through the nucleosome. The data suggest that, as expected, PARP1 E988K cannot be automodified even in the presence of NAD+ [41], and cannot PARylate proteins facilitating transcription through the nucleosome, although it can still form complexes with nucleosomes that result in partial inhibition of transcription. Thus, the catalytic function of PARP1 is required for its positive effect on transcription through the nucleosome in the presence of NAD+. 

To further evaluate the role of PARP1-induced PARylation in yPol II transcription through the nucleosome, transcription was conducted in the presence of PARG—an enzyme that strongly reverses PARylation by PARP1 (Appendix A). The outcomes of transcription in the presence of PARP1/NAD+/PARG and transcription in the absence of PARP1 are indistinguishable (Figure 2c), suggesting that PARylation is essential for PARP1-dependent transcription through the nucleosome. Note that in the presence of PARP1/NAD+/PARG, transcription through the nucleosome is not inhibited (as was observed in the presence of PARP1 alone), suggesting that PARG (Appendix A) could partially prevent formation of PARP1-nucleosome complexes that inhibit the transcription.

Taken together, the data suggest that in the absence of NAD+, PARP1 moderately inhibits transcription through the nucleosome, likely forming PARP1-nucleosome complexes [23,40,42]. In the presence of NAD+, PARP1 catalyzes auto-PARylation of itself and possibly other protein (s) that results in disruption of PARP1-nucleosome complexes [38,39,40] and facilitates transcription through the nucleosome. PARG reverses PARP1-induced PARylation and PARP1-dependent transcription through the nucleosome.

### 2.4. PARP1-Induced PARylation Facilitates Formation of Productive Elongation Complexes

The pattern of nucleosome-specific pausing is dictated by formation of various intermediates containing nucleosome with perturbed structure during transcription [28,43]. Therefore, quantitative, time-resolved analysis of the effect of PARP1 on the pausing can help to identify the intermediates affected by this protein, as was shown for another histone chaperone FACT [35,44]. Transcription through a nucleosome is a complicated multi-step process that cannot be described by a simple set of kinetic equations; therefore, KinTek Kinetic Explorer software [45] that previously allowed analysis of a complicated rate of similar, FACT-dependent transcription through a nucleosome [35,44] was employed. 

To study the effect of PARP1 on the pausing, time-courses of transcription through the 603 nucleosome in the presence or absence of PARP1 were analyzed (Figure 3a). yPol II pausing was quantified, the data plotted against time and fit to the sequential multi-step model (Figure 3b) that was developed and validated previously for analysis of the effect of FACT on transcription through the nucleosome [35,44]. Since during transcription through the nucleosome by yPol II, both productive and non-productive (stalled and arrested) ECs are formed [43,46,47], reversible formation of non-productive complexes at each step during the elongation is incorporated in the model (Figure 3b). In the case of FACT-dependent transcription it has been established that this is a minimal model that produces a good fit to the experimental data [35,44]. Transcription through a nucleosome is a highly conservative process that proceeds through similar intermediates formed in the presence of FACT [35,44] or PARP1 (Figure 3a). Therefore, the same model was used for analysis of the effect of PAPR1 on transcription through the nucleosome.

The data show an excellent fit (*p* value < 0.01) of the entire set to the sequential model using KinTek software (Appendix A). The KinTek software allowed us to choose well-constrained and 95%-confident rate parameters. Specifically, in the presence of PARP1/NAD+ the rates of conversion from productive to non-productive complexes are strongly (up to >2-fold) decreased, and the reverse rates are increased (Figure 3c). Thus, PARP1 inhibits formation of non-productive and promotes formation of productive ECs and thereby facilitates transcription through the nucleosome. 

Our previous analysis of the structures of the intermediates formed during transcription through the nucleosome by yPol II [28] allows approximate assignment of the kinetic steps to corresponding structural changes (Figure 3d). The effect of PARylation on the rates is maximal during transcription through the central +(35–65) region of nucleosomal DNA. This region of nucleosomal DNA interacts with H3/H4 histone tetramer; the nucleosomal DNA is maximally uncoiled from the histone octamer during yPol II transcription through this region (Figure 3d). Accordingly, the data suggest that negatively charged PARP1-PARylated protein (s) preferably interact with the H3/H4 tetramer surface exposed during transcription. Alternatively, the PARylated protein (s) interact with the positively charged octamer surface formed by both H2A/H2B dimers and H3/H4 tetramers that is more extensively open during transcription through the +(35–65) region. The second model is preferred because PARylated PARP1 positively, although to a lower degree, affects transcription through the +(5–35) region where the surface of H3/H4 tetramer is not exposed (Figure 3c).

### 2.5. PARP1 Facilitates Histone Eviction during Transcription

Since negatively charged PARylated protein (s) that facilitate transcription through the nucleosome likely interact with the positively charged octamer surface exposed during transcription, it is expected to compete with the DNA-histone interactions in the nucleosome, and thereby destabilize transcribed chromatin, and possibly induce histone eviction. 

To analyze the efficiency of histone eviction during transcription, 603 nucleosomes were DNA end-labeled, transcribed in the absence or presence of PARP1/NAD+, and the products were analyzed by native PAGE (Figure 4a). Transcription in the presence of PARP1 results in a strong increase in the amount of released DNA (Figure 4b) and corresponding decrease in the amount of the nucleosomes after transcription by yPol II (Figure 4a). Appearance of a larger amount of histone-free DNA after transcription indicates that more core histones were evicted from DNA in the presence of PARP1/NAD+. As expected, PARP1 does not induce histone eviction from DNA in the absence of transcription (Appendix A). These observations are consistent with the data obtained in vivo in Drosophila where PARP1 induces chromatin decondensation and histone loss [25,48].

### 2.6. Anti-Cancer Drug Olaparib Strongly Inhibits PARP1-Dependent Transcription through a Nucleosome

Since the catalytic activity of PARP1 is essential for its effect on transcription through the nucleosome, it is expected that clinical PARP1 inhibitors affecting the active center of the enzyme (e.g., FDA-approved olaparib [13,14]) would be inhibitory for the transcription. To evaluate the effect of olaparib on PARP1-dependent transcription through the nucleosome, PARP1 was pre-incubated with the inhibitor and transcription was conducted in the presence of PARP1/olaparib (Figure 5). Clearly, olaparib strongly inhibits PARP1-, NAD+-dependent transcription through the nucleosome. Furthermore, nucleosomal pausing at the +45 region that is magnified after binding of PARP1 to the nucleosome in the absence of NAD+ (Figure 1) is further increased in the presence of olaparib, likely reflecting trapping of PARP1-nucleosome complex induced by this inhibitor [49,50].

## 3. Discussion

In summary, our data show that human PARP1 strongly facilitates transcription through the nucleosome in the presence of NAD+ (Figure 1), and its catalytic activity is essential for this activity (Figure 2). Catalytically defective PARP1 (E988K) and intact PARP1 in the absence of NAD+ moderately inhibit transcription through the nucleosome (Figure 1 and Figure 2). PARP1-mediated PARylation facilitates transcription through the nucleosome by decreasing the rates of conversion from productive to non-productive complexes and increasing the reverse rates (Figure 3); the positive effect of PARP1 on transcription is particularly strong when yPol II progresses through the +(35–55) region of nucleosomal DNA (Figure 3c). Transcription in the presence of PARP1/NAD+ induces histone eviction on a fraction of nucleosomal templates (Figure 4). Finally, anti-cancer drug olaparib strongly inhibits PARP1-dependent transcription through a nucleosome (Figure 5). 

### 3.1. Mechanism of PARP1 Action during Transcription through a Nucleosome

Previously, we and others have shown that non-PARylated PARP1 can interact with nucleosomes [38,39,40] and change the nucleosomal structure [38]. Apparently, PARP1-nucleosome interactions make transient uncoiling of nucleosomal DNA from the histone octamer by transcribing Pol II (see Figure 6, pathway 1–4) more difficult, thus inducing a stronger Pol II pausing during transcription in the presence of PARP1 without NAD+ (Figure 1b). This effect is observed during transcription through the +(25–40) region of nucleosomal DNA, after uncoiling of at least 35–50 bp of nucleosomal DNA by Pol II [28], suggesting that at least some PARP1-nucleosome interactions persist after nucleosomal structure is transiently disturbed during transcription.

In contrast, when NAD+ and PARP1 are present in the reaction, transcription through the nucleosome by Pol II is strongly facilitated. Since catalytic activity of PARP1 is essential for this activity, PARP1-dependent PARylation of one or several proteins present in the reaction (PARP1, core histones [9,37] and Pol II [37]) likely facilitates transcription. Transcription through a nucleosome involves sequential, partial and reversible uncoiling of nucleosomal DNA from the octamer accompanied by disruption and re-formation of multiple electrostatic DNA-histone interactions (Figure 6, pathway 1 to 4 [28,43]). PAR carries a strong negative charge and, therefore, could form complexes with the positively charged DNA-binding surfaces of the histone octamer that are transiently exposed during transcription. Thus, PARylated protein(s) likely interfere with re-formation of DNA-histone interactions by forming competitive PAR-histone complexes. Formation of PAR-histone complexes would prevent DNA re-coiling on the histone octamer during transcription and facilitate further DNA uncoiling from the octamer and transcription through the nucleosome (Figure 6, pathway 1′ to 4′). In particular, PARylated protein(s) are expected to interact with the exposed DNA-binding surfaces of core histones behind the transcribing enzyme (Figure 6, intermediates 2′ and 3′) that otherwise allow formation of a small intranucleosomal DNA loop that facilitates nucleosome survival during transcription [28,34]. Thus, PARylated PARP1 compromises formation of the primary intermediate that facilitates nucleosome survival during transcription and induces partial histone eviction by transcribing Pol II (Figure 4).

The PARylated proteins most efficiently facilitate transcription through the +(35–55) region of nucleosomal DNA (Figure 3), where the most “open” intermediates containing DNA uncoiled from H3/H4 tetramer are formed (Figure 6, pathway 1–4). The data suggest that the negatively charged PARylated protein (s) either interact with the positively charged surface of histone H3/H4 tetramer or require a large surface of the histone octamer, including both H2A/H2B dimer and H3/H4 tetramer surfaces available during transcription through the +(35–55) region (Figure 4).

As we have shown previously [38,51], in the absence of transcription, PARP1 reversibly affects the nucleosome structure (Appendix A). In an apparently contradicting study, it has been shown previously that PARP1 binding can induce DNA dissociation from nucleosomes in vitro [9]. Although the reason for the controversy is unknown, in several other studies, it has been shown previously that PARP1 can disrupt higher order chromatin structure with a minimal effect on nucleosome organization [27,40,42]. 

### 3.2. Mechanism of Olaparib Action during Transcription through a Nucleosome

Olaparib is a competitive inhibitor of the catalytic center of PARP proteins [52], and is expected to inhibit PARP1-dependent PARylation and, in turn, inhibit transcription activated by protein PARylation. Indeed, olaparib strongly inhibits PARP1-, NAD+-dependent transcription through the nucleosome (Figure 5). We have also shown that olaparib can trap PARP1-nucleosome complexes even in the presence of NAD+, which in the absence of olaparib induces dissociation of PARP1 from nucleosomes [38]. This observation explains why PARP1-dependent pausing is maintained and slightly magnified in the presence of NAD+ and olaparib (Figure 5): apparently, PARP1 trapped in the complex with the nucleosome moderately inhibits progression of yPol II. The data suggest that olaparib can target PARP1 involved in both DNA repair and gene regulation, as was suggested previously [53].

### 3.3. Implications In Vivo

Our in vitro data suggest the following scenario for PARP1 action on transcribed genes. PARP1 is associated with nucleosomes at the promoters, replacing linker histone H1 [22,27,54] and is capable of ADP-ribosylating various proteins/protein complexes in the vicinity upon gene activation. As a gene is activated, PARP1 could modify NELF-E protein [20], itself, core histones and Pol II [9,37], and thereby affect transcription through chromatin, especially through the +1 nucleosome. PARP1-modified proteins/protein complexes facilitate transcription through the +1 nucleosome and possibly other nucleosomes on the gene; indeed, hPARP1 is present on transcribed regions of active genes [21,54,55,56]. PARP1 could also increase the rate of histone turnover and possibly induce histone eviction from transcribed genes, transiently disrupting at least the +1 nucleosome. Olaparib can trap PARP1 in the PARP1-nucleosome complex and moderately inhibit transcript elongation.

### 3.4. Conclusions

Our data suggest that the enzymatic activity of PARP1 is essential for more efficient overcoming of the nucleosomal barrier to transcript elongation by Pol II and for nucleosome survival during this process. During transcription, one or some of the proteins participating in transcript elongation through a nucleosome are poly(ADP-ribosyl)ated by PARP1 (PARP1 itself and/or Pol II), thus acquiring a negative charge. The negatively charged surfaces of the protein (s) likely interact with positively charged surfaces of other proteins (most likely core histones) that are normally involved in formation of the nucleosomal barrier to transcription through their interaction with nucleosomal DNA. As a result, the histone-DNA interactions are weakened, the height of the nucleosomal barrier is decreased and Pol II can more efficiently traverse the nucleosome. This function of PARP1 is probably essential for transcription through the +1 nucleosome and possibly other nucleosomes on a gene in vivo.

## 4. Materials and Methods

### 4.1. Purification of Proteins, DNA Templates, and Nucleosomes

Yeast Pol II [57], human Pol II [58], human PARP1 protein [7] and histones [43] were purified as described. Nucleosomes were assembled on the 227–bp 603 nucleosome positioning sequence using salt gradient method, as previously described [28]. DNA templates for nucleosome assembly were synthesized by polymerase chain reaction (PCR), digested with *Tsp*R1 and purified by a gel extraction kit (Omega BioTek, Norcross, GA, USA). Then, DNA templates were mixed with purified chicken erythrocyte H2A/H2B histones dimers and H3/H4 histones tetramers at a 1:1.8:1.2 molar ratio in the presence of shredded salmon sperm DNA (in three-fold weight excess over DNA templates) in the following buffer: 2 M NaCl, 10 mM Tris-HCl (pH 7.4), 0.1% NP-40, and 0.2 mM EDTA (pH 8). The mixture of DNA and histones was then dialyzed at 4 °C against set of buffers containing 10 mM Tris-HCl (pH 7.4), 0.1% NP-40, and 0.2 mM EDTA (pH 8.0), and stepwise decreasing concentrations of NaCl (2, 1.5, 1, 0.75, 0.5 M, and 10 mM), as described [59]. Assembled nucleosomes were analyzed on 6% native poly-acrylamide gel (39:1) in 0.5xTBE. 

### 4.2. Transcription and Nucleosome Fate Assays

Yeast Pol II elongation complexes were assembled on a 50-bp DNA fragment by incubation of nine-nucleotide RNA fragment, yPol II and 59-bp non-template strand as described [59]. These oligonucleotides are complementary and form *Tsp*R1 site at one end after annealing. The yPol II elongation complexes were immobilized on nickel-NTA-agarose beads (Qiagen, Hilden, Germany), washed, eluted from the beads, and ligated to the nucleosomes. yPol II was allowed to advance to the -5 position relative to the nucleosomal boundary using [α-^32^P] NTPs to pulse-label the RNA. Reactions were then incubated at 22 °C in the absence or presence of 100 nM PARP1 and 100 μM NAD+ for 15 min. Transcription was then resumed by the addition of a large excess of unlabeled NTPs in the presence of 150 mM KCl for 10 min. Samples were then analyzed on 8% denaturing poly-acrylamide gel (19:1), and the bands were scanned on Pharos FX Plus Molecular Imager (BioRad, Hercules, CA, USA).

For nucleosome fate experiments, the non-template 59-bp DNA strand was end-labeled by [γ-^32^P] ATP prior to the assembly of elongation complexes, and transcription was carried out in the presence of unlabeled NTPs at 150 mM KCl, as described above for transcription experiments. Samples were analyzed on 4% native poly-acrylamide gel (39:1) in 0.5× TBE. Amount of histone-free DNA was then calculated as a fraction of transcribed templates using OptiQuant Software, Version 04.00 (PerkinElmer, Shelton, CT, USA) [59]. There are two primary products of transcription: DNA and nucleosomes that originate from fully transcribed template. Therefore, the fraction of free DNA originating from fully transcribed templates was calculated as DNA, % = DNA/(DNA + nucleosomes). The background values for DNA and nucleosomes present in the absence of complete transcription (-U reaction, Figure 4a, first lane) were subtracted. 

Transcription by human Pol II was performed, as described for yeast Pol II above, with minor modifications. In brief, hPol II ECs assembled using 3′ end-biotinylated template strand DNA oligonucleotide were immobilized on streptavidin magnetic beads in TB40 (20 mM Tris-HCl, pH 7.9, 5 mM MgCl2, 40 mM KCl, 1 mM β-mercaptoethanol) for 10 min and then washed once with TB300 and twice with TB40. Pre-formed hPol II ECs were ligated to reconstituted nucleosomes. Ligated products were first incubated in the presence of ATP, CTP, and [α-32P] GTP to advance the polymerase to the -5 position and were then incubated in the presence of all NTPs in TB150 for 10 min at RT. 

### 4.3. Time-Courses of yPol II Transcription

Time-courses of the transcription through nucleosome were measured for indicated time intervals at 22 °C as described in [35,44]. The yields of the RNA products at each time point (3, 15, 45 and 120 s) were quantified using OptiQuant Software, Version 04.00 (PerkinElmer, Shelton, CT, USA) and computationally fitted to the kinetic model using the KinTek Explorer software, version 4(Snow Shoe, PA, USA) [45]. All parameters were obtained after the simulation. The *p* value (the quality of data fitting in our model) was calculated by the software. Average rate constants and standard deviations were calculated from three independent experiments. 

### 4.4. PARP1 Automodification

Transcription reactions were performed as described above. The proteins were then separated on 4–12% Novex PAAG (Thermo Fisher, Waltham, MA, USA) and transferred to PVDF membrane. The membrane was blocked with PBS-T +5% milk and then incubated with primary anti-PAR monoclonal antibodies (Santa Cruz Biotechnology, Dallas, TX, USA SC-56198) and anti-PARP1 polyclonal antibodies (Abcam, Cambridge, United Kingdom ab227244). After incubation with secondary HRP-conjugated antibodies signals were developed by incubation of the membrane with SurerSignal West Femto Maximum Sensitivity Substrate (Thermo Fisher, Waltham, MA, USA). 

### 4.5. Core Histones ADP-Ribosylation

Nucleosomes were incubated in the transcription buffer in the presence or absence of PARP-1 and/or bio-NAD+ at room temperature for 25 min. The proteins were then separated on 4–12% Novex PAAG (Thermo Fisher, Waltham, MA, USA), stained with Sypro Orange and transferred to PVDF membrane. The membrane was blocked with PBS-T +5% milk and then incubated with horse radish peroxidase (HRP)-conjugated streptavidin. Signals were developed after incubation of the membrane with SurerSignal West Femto Maximum Sensitivity Substrate (Thermo Fisher, Waltham, MA, USA). 

## Figures and Tables

**Figure 1 ijms-23-07107-f001:**
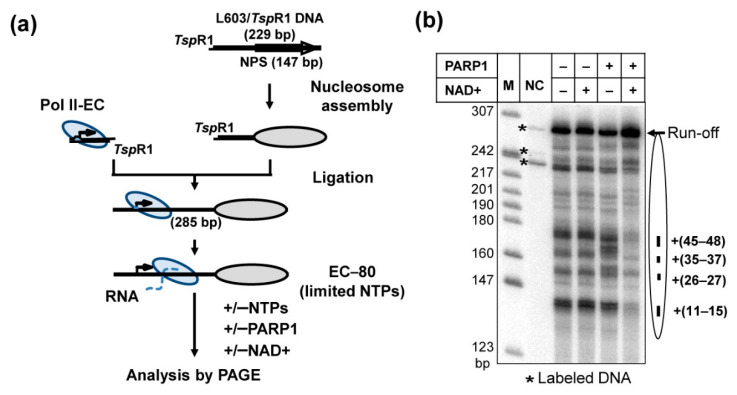
Experimental approach for analysis of the effect of PARP1 on transcription through a nucleosome. (**a**) In vitro transcription assay for analysis of the effect of PARP1 on transcription through a nucleosome by yeast Pol II. yPol II elongation complex (yPol II-EC, the arrow indicates direction of transcription) was assembled and ligated to the nucleosome assembled on the 603 nucleosome positioning sequence (NPS) via TspRI ends. EC-80 was formed in the presence of ATP, CTP, and [α-^32^P] GTP (-UTP reaction [36]). The numerical index of EC indicates the position of the yPol II active center relative to the promoter–proximal nucleosomal boundary. Then, transcription through the nucleosome was continued in the presence of an excess of all unlabeled NTPs and/or PARP1 and NAD+. (**b**) PARP1 facilitates transcription through 603 nucleosome in the presence of NAD+. Transcription through the nucleosome was conducted in the absence or presence of PARP1 and NAD+. Analysis of pulse-labeled RNA by denaturing PAGE. Positions of the nucleosome and the major pausing sites are indicated on the right. Labeled DNA fragments are indicated by asterisks. NC—no chase. M: end-labeled pBR322-MspI digest.

**Figure 2 ijms-23-07107-f002:**
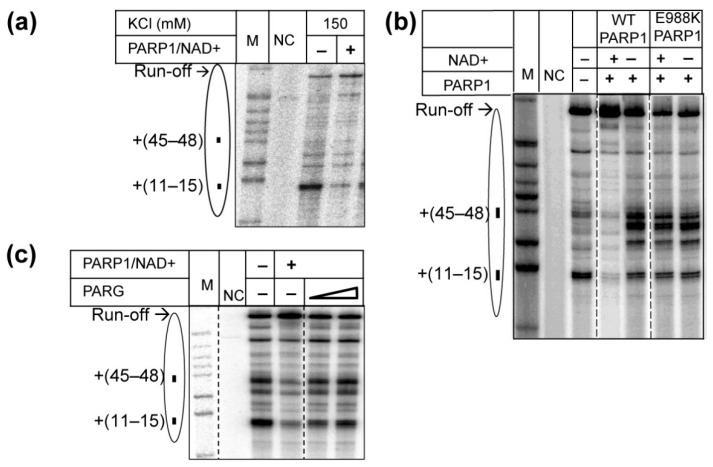
Catalytic activity of PARP1 is required to facilitate Pol II transcription through a nucleosome. (**a**) PARP1 facilitates transcription through the nucleosome by human Pol II. hPol II elongation complex was assembled and ligated to the 603. Transcription was conducted in the presence or in the absence of PARP1 and NAD+, as described in Figure 1a. (**b**) Catalytic activity of PARP1 is required to facilitate yPol II transcription through the nucleosome. Transcription through the nucleosome was conducted by yPol II as described in Figure 1a. (**c**) The effect of PARP1 on transcription through the nucleosome is compromised by PARP1 antagonist PARG. Analysis of pulse-labeled RNA by denaturing PAGE. Other designations are the same as in Figure 1.

**Figure 3 ijms-23-07107-f003:**
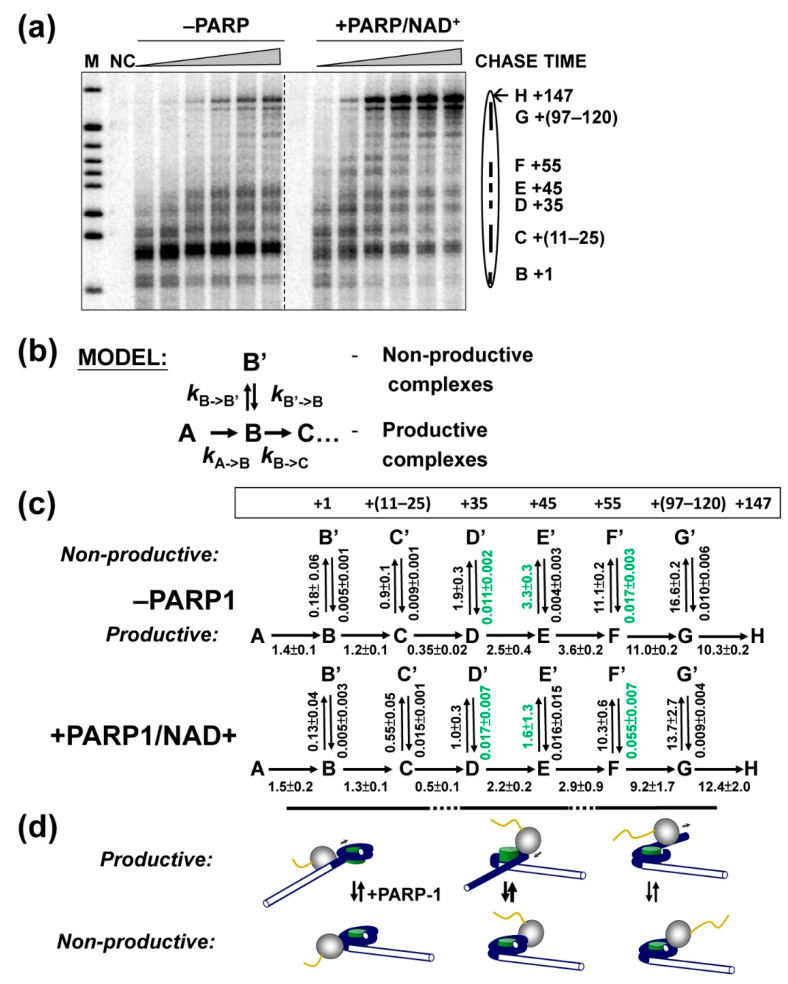
PARP1 facilitates formation of productive intranucleosomal yPol II complexes. (**a**) 603 nucleosomes were transcribed by yPol II at 150 mM KCl in the absence or presence of PARP1/NAD+ for different time intervals (3, 10, 45, 120, 300 or 600 s), followed by analysis of pulse-labeled RNA by denaturing PAGE. The intranucleosomal pauses (from B to H) were quantified using Pharos FX Plus Molecular Imager (BioRad). (**b**) The quantified data were analyzed using an elongation model that produces a good fit of the experimental data to the calculated curves [35]. The model assumes that at every position of yPol II on the nucleosomal DNA, there is a probability to form both productive complexes (B to H) and reversible non-productive complexes (B’ to H’). (**c**) KinTek Kinetic Explorer software [45] was used to determine the rate constants (μM^−1^ s^−1^) for each step of transcription through the nucleosome. Averages from three experiments and the standard deviations are shown. The rate constants positively affected by PARP1 by more than two-fold are shown in green. (**d**) The expected complexes formed at each region [28] are shown.

**Figure 4 ijms-23-07107-f004:**
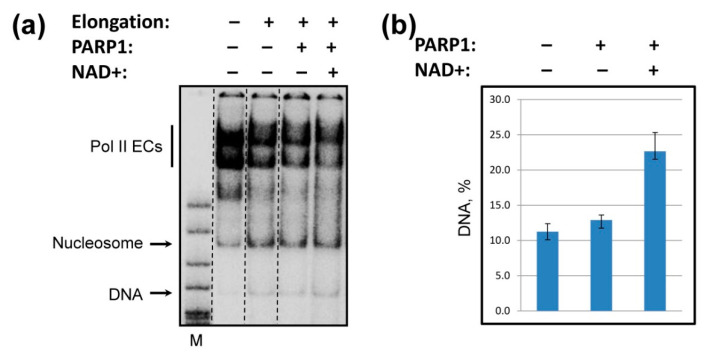
PARP1 facilitates histone eviction during transcription. (**a**) DNA-end-labeled 603 nucleosomes were transcribed in the presence or in the absence of PARP1 and/or NAD+ at 150 mM KCl as described in Figure 1a and analyzed by native PAGE. (**b**) The amounts of histone-free DNA produced after transcription was quantified (presented as a fraction of transcribed templates, see Methods). Averages and standard deviations from three independent experiments are shown. During transcription in the absence of PARP1, a fraction of templates (~10%) lose histone octamer [28]. In the presence of PARP1/NAD+ the amount of histone-free DNA produced during transcription by yPol II is considerably increased.

**Figure 5 ijms-23-07107-f005:**
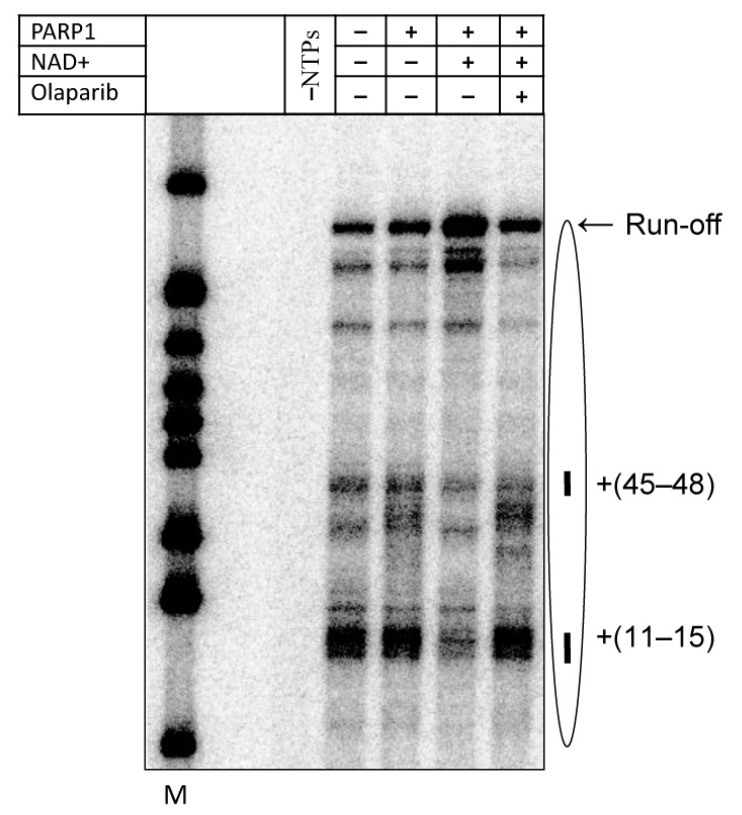
Anti-cancer drug olaparib strongly inhibits PARP1-dependent transcription through a nucleosome. PARP1 was pre-incubated in the presence of olaparib and added to pre-formed yPol II elongation complexes EC-80. Then 603 nucleosomes were transcribed in the presence or in the absence of PARP1 and/or NAD+, as described in Figure 1a,b.

**Figure 6 ijms-23-07107-f006:**
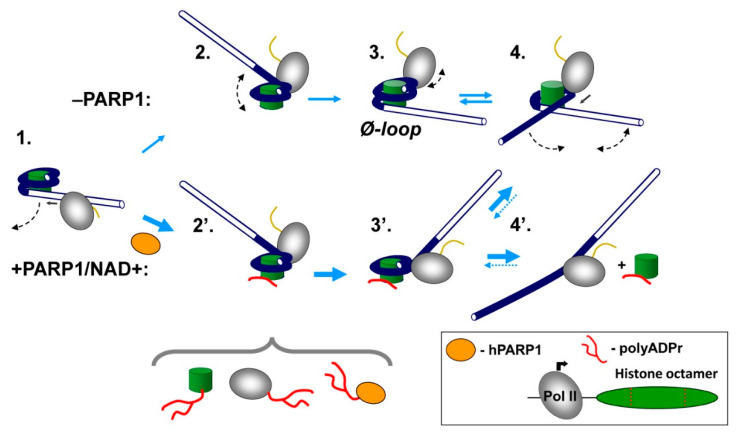
Proposed mechanisms of PARP1-dependent transcription through a nucleosome. Pol II transcription through the nucleosome in the absence of PARP1 (complexes 1 to 4 [28]) is accompanied by initial partial uncoiling of nucleosomal DNA from the octamer in front of Pol II (complex 2), re-coiling of nucleosomal DNA behind the transcribing enzyme with formation of a small transient intranucleosomal DNA loop (Ø-loop, complex 3), and subsequent partial uncoiling of nucleosomal DNA in front of the enzyme (complex 4). Intermediates 2 to 4 are formed during transcription through the +(35–55) region of nucleosomal DNA [28,43] that is most affected in the presence of PARP1/NAD+. Since PARP1 could PARylate itself and each protein complex present during transcription in vitro (core histones and Pol II [9,37]), three mutually non-exclusive scenarios are possible. In each of them negatively charged PAR attached to a protein interacts with transiently exposed, positively charged surface of the histone octamer, preventing the re-coiling of nucleosomal DNA and formation of the intranucleosomal DNA loop (complexes 2′ and 3′). As a result, in the presence of NAD+ and PARP1, the PARylated protein(s) facilitate transcription through the nucleosome and histone eviction (complex 4′).

## Data Availability

The data presented in this study are available on request from the corresponding author.

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
