# Peer review of "Human PARP1 Facilitates Transcription through a Nucleosome and Histone Displacement by Pol II In Vitro"

_ijms, 2022, doi:10.3390/ijms23137107_

Round 1

Reviewer 1 Report

Please, see attached file.

Reviewer 2 Report

In this manuscript Kotova et al examine the effect of PARP1 on in vitro elongation of yeast or human Pol II and find that PARP1 catalytic activity promotes transcription through a nucleosome by overcoming pause sites. PARP1 mono-ADPR mutant and PARP1 inhibitor slightly impaired transcription efficiency and reinforced pausing, possibly due to the formation of PARP1-nucleosome complexes that impede Pol II traversal through the nucleosomes. The authors modelled their data to show that PARP1 promotes the formation of productive and reduces the formation of non-productive elongation complexes. Lastly, the authors show that PARP1 activity induces histone eviction from nucleosomes as a likely mechanism that facilitates Pol II traversal through the nucleosomes. The experiments were well-executed, the conclusions are sound and the manuscript is well-written. I have two suggestions on how to improve the manuscript.

Major:

Chromatin remodelers such as CHD1 and the histone chaperone complex FACT are known to facilitate Pol II transcription through chromatin. Thus I suggest to test how PARP1 affects transcription in their presence. This would make the study more functionally relevant as it would bring it one step closer to the nuclear context.

In this manuscript Kotova et al examine the effect of PARP1 on in vitro elongation of yeast or human Pol II and find that PARP1 catalytic activity promotes transcription through a nucleosome by overcoming pause sites. PARP1 mono-ADPR mutant and PARP1 inhibitor slightly impaired transcription efficiency and reinforced pausing, possibly due to the formation of PARP1-nucleosome complexes that impede Pol II traversal through the nucleosomes. The authors modelled their data to show that PARP1 promotes the formation of productive and reduces the formation of non-productive elongation complexes. Lastly, the authors show that PARP1 activity induces histone eviction from nucleosomes as a likely mechanism that facilitates Pol II traversal through the nucleosomes. The experiments were well-executed, the conclusions are sound and the manuscript is well-written. I have two suggestions on how to improve the manuscript.

Major:

Chromatin remodelers such as CHD1 and the histone chaperone complex FACT are known to facilitate Pol II transcription through chromatin. Thus I suggest to test how PARP1 affects transcription in their presence. This would make the study more functionally relevant as it would bring it one step closer to the nuclear context.

Minor:

The scheme in Fig. 3b is hard to read and interpret. Please improve this.

Reviewer 3 Report

Kotova et al studied how human PARP1 protein affect in vitro transcription by yeast and human Pol II with +1 nucleosome DNA. They show the kinetic effects of addition of PARP1 only vs. PARP1+NAD+, and tested how the PARP1 inhibitor olaparib modulates transcription activities. Authors then proposed the mechanisms by which PARP1 facilitate transcription through a nucleosome.

Major concerns: this work employs kinetic approaches yet the analysis was not sound.

1. Fig. 3b, authors should provide detailed kinetics models, with equations, for others to really understand and evaluate the fitting by the non-open access KinTek software. Line 196, what is an excellent fit?

2. Fig. 3c, what is the unit for the listed numbers? Errors of the fitting? How are errors propagated when averaging the three replicates?

3. Fig. 4a, it doesn't seem like the intensity of the +PARP1+NAD+ DNA band was twice the intensity of the DNA band in -PARP1-NAD+, raising a question to the bar plots in Fig. 4b.

4. Fig. S4, are authors sure that the color code is correct now? Can authors use a log10 x-axis to make the plots more readable? And please describe the axes, too.

Minor points:

1. Fig. 6, can authors test their models by site-specific restriction enzyme digestion of the DNA?

2. Fig. S1 and its original gel did not seem to match, they were different gels.

3. Fig. S2, authors should show a blot against PARP1 protein, so we could understand whether the anti-PAR signal was from PARP1 or other components in the transcription reaction, which is a key hypothesis in this manuscript. Authors should also simply mix PARP1 and NAD+ in the transcription buffer and see if there is auto-PARylation, which would also help understanding the mode of PARP1 function during transcription with nucleosome.

4. Fig. S3, it seems like at higher KCl the run-off products built the most. Why didn't authors choose to perform transcription using this more efficient condition?

5. Authors should include all the gel pictures from their three replicates in the supplementary materials. This is key to the paper, as many of the conclusions were based on comparing the subtle differences in band intensities from different conditions. Reproducibility is thus critical.

Round 2

Reviewer 1 Report

Please, see attached file.

Author Response

Reviewer 1

The authors have satisfactorily responded to my questions and made the necessary changes to the manuscript, and now I had only minor comments:

  1. I think that “PARP1 and NAD+” at the ends of the sentence “Then the templates were incubated in the absence or presence of hPARP1 and NAD+, and transcribed in the presence of an excess of all unlabeled NTPs, PARP1 and NAD+” should be deleted.

Reply: Deleted as requested.

  1. Please, use uniform designation for PARP1 (hPARP1 or PARP) in the text.

Reply: Designations for PARP1 were unified.

  1. Please, correct [α-32P] GTP (line 94), “32” should have superscript format.

Reply: Corrected.

Reviewer 2 Report

I am looking forward to your future results with PARP1 and elongation factors!

Author Response

Thank you!

Reviewer 3 Report

1. Authors tried to address my concerns about their kinetic fitting, yet the answer was not satisfactory.

First, the technical quality of this manuscript should not be supported by a previous report with ~20 citations/year, but the data analyses in this work. Anyway, let's take a look at the Fig. S10 of the well accepted PNAS paper from the authors and their collaborators, which is of the same type of the new Fig. S5 in this manuscript (key kinetic analysis here). Top pannel, -FACT, are we seeing two curves with the same green color (the reviewer is supposed to be not color blinded), which are B and G? Is the appearance of intermediate E occurring prior to intermediate D and how could this be even possible considering the stepwise reaction scheme? Why are we missing intermediates H and J in the fitting but they were in the kinetic models presented in Fig. 3 of the PNAS paper? Bottom pannel has the same problems. If such analyses were the only support of the integrity of the current work, this work should never be considered solid.

Second, let's now look at the new Fig. S5 in this revised version. The axes were still poorly labeled, especially the y-axis numbers were hidden by the curves. More importantly, authors claimed that the color code was and is correct. If we look at the red curve in the left pannel, annotated as B by authors, which reached to <10% at timepoint 120s. However, if we look at the original gel in Fig. 3a left pannel, intermediate B is clearly the most abundant species in all the products, and even by eye should be much more than 10% of total. 

Third, authors gave a p value <0.01 to support their "excellent fit" from so limited data points. What does the p value even mean here? Perhaps the reviewer is just too dumb to understand their statistical analyses. Also, the errors reported by the authors in the new Fig. 3c were mostly less than 10%, sometimes even less than 5%, how could that be if the data points were mostly outside the curves in Fig. S5? Even just fitting a single curve the errors would be large, not to say the global fitting of 7 curves together with only 4 data points per curve. Authors, please take a full screen shot of the KinTek (which version?) analysis from each replicate and put them in the supplementary files (including each rate and error), so others can assess if the model fits.

2. It is very weird how authors cut and stitched their Fig. 4a, which by eye looked as if those lanes were not artificially put together, and therefore super misleading. This must be corrected with separate lanes. Same issue may be associated with other figures, too.

Author Response

Reviewer 3

  1. Authors tried to address my concerns about their kinetic fitting, yet the answer was not satisfactory.

First, the technical quality of this manuscript should not be supported by a previous report with ~20 citations/year, but the data analyses in this work. Anyway, let's take a look at the Fig. S10 of the well accepted PNAS paper from the authors and their collaborators, which is of the same type of the new Fig. S5 in this manuscript (key kinetic analysis here). Top pannel, -FACT, are we seeing two curves with the same green color (the reviewer is supposed to be not color blinded), which are B and G? Is the appearance of intermediate E occurring prior to intermediate D and how could this be even possible considering the stepwise reaction scheme? Why are we missing intermediates H and J in the fitting but they were in the kinetic models presented in Fig. 3 of the PNAS paper? Bottom pannel has the same problems. If such analyses were the only support of the integrity of the current work, this work should never be considered solid.

Reply: We hope to avoid these mistakes in the current work.

Second, let's now look at the new Fig. S5 in this revised version. The axes were still poorly labeled, especially the y-axis numbers were hidden by the curves. More importantly, authors claimed that the color code was and is correct. If we look at the red curve in the left pannel, annotated as B by authors, which reached to <10% at timepoint 120s. However, if we look at the original gel in Fig. 3a left pannel, intermediate B is clearly the most abundant species in all the products, and even by eye should be much more than 10% of total. 

Reply: We thank the reviewer for the important comments. Indeed, in the previous version of the manuscript some bands in Fig. 3a were mislabeled. The labeling of the bands has been corrected and now is in correspondence with the Fig. S5.

Third, authors gave a p value <0.01 to support their "excellent fit" from so limited data points. What does the p value even mean here? Perhaps the reviewer is just too dumb to understand their statistical analyses. Also, the errors reported by the authors in the new Fig. 3c were mostly less than 10%, sometimes even less than 5%, how could that be if the data points were mostly outside the curves in Fig. S5? Even just fitting a single curve the errors would be large, not to say the global fitting of 7 curves together with only 4 data points per curve. Authors, please take a full screen shot of the KinTek (which version?) analysis from each replicate and put them in the supplementary files (including each rate and error), so others can assess if the model fits.

Reply: We have added a new table accompanying the Fig. 3c with KinTek raw data and the calculations. Unfortunately, we did not keep all individual screen shots of the KinTek.

The p value here means the quality of data fitting in our model; it was calculated by the software (now clarified in the Materials and Methods).

KinTek Explorer software, version 4 was used (now clarified in the Materials and Methods).

  1. It is very weird how authors cut and stitched their Fig. 4a, which by eye looked as if those lanes were not artificially put together, and therefore super misleading. This must be corrected with separate lanes. Same issue may be associated with other figures, too.

Reply: In the Figs. 3a and 4a the stitched lanes are now separated by dashed lines. Please accept our apologies for forgetting to add the separating lines previously.

Round 3

Reviewer 3 Report

Not sure if I missed it, but I could not find the table accompanying Fig 3c in the new files downloaded.

Now the new Fig. S5 has better Y axis labels. However, authors really did not fix the problem, and seem still confused by the data. What is annotated in Fig.S5 as the red dots/curves is intermediate B. With the addition of an intermediate A to the new Fig.3a, intermediate B in the left pannel still corresponds to the most abundant one across all the timepoints, thus should look more like the green curve in the left pannel of Fig. S5.

Nevertheless, I feel strongly that the authors should consult their kinetics experts, and carefully re-exam all the raw gels to better annotate the different species detected by the gels at different timepoints. One thing must be fixed is adding proper labeling of the marker lanes to all the gels. By eye, what is now annotated as B (+5) in the new Fig.3a would correspond to the +(11-15) in those gels in Figures 1 and 2. These also show that the authors poorly understood the gels/assays, and how could the conclusions made be convincing to readers?

Author Response

Reply to the Editor’s/reviewers’ comments to IJMS manuscript ijms-1733323 “Human PARP1 Facilitates Transcription through Chromatin and Histone Displacement by Pol II in vitro

Apparently, the reviewer 3 was reviewing an older version of the manuscript (see below).

Reviewer 3

Not sure if I missed it, but I could not find the table accompanying Fig 3c in the new files downloaded.

Reply: The table accompanying Fig 3c was included in the pdf file “ijms-1733323revised2 Raw data 061122” that was submitted with the revised manuscript, as was described in the accompanying letter. Perhaps the reviewer was looking at an older version of the Raw data file.

Now the new Fig. S5 has better Y axis labels. However, authors really did not fix the problem, and seem still confused by the data. What is annotated in Fig.S5 as the red dots/curves is intermediate B. With the addition of an intermediate A to the new Fig.3a, intermediate B in the left pannel still corresponds to the most abundant one across all the timepoints, thus should look more like the green curve in the left pannel of Fig. S5.

Reply: There is no intermediate A in the new Fig.3a in the revised manuscript. Therefore, it seems that the reviewer was looking at the last submitted version of the Supplemental Figures file (submitted as “ijms-1733323revised2 SUPPL FIGS 061122”) but at an older version of the manuscript (see below).

Nevertheless, I feel strongly that the authors should consult their kinetics experts, and carefully re-exam all the raw gels to better annotate the different species detected by the gels at different timepoints. One thing must be fixed is adding proper labeling of the marker lanes to all the gels. By eye, what is now annotated as B (+5) in the new Fig. 3a would correspond to the +(11-15) in those gels in Figures 1 and 2. These also show that the authors poorly understood the gels/assays, and how could the conclusions made be convincing to readers?

Reply: There is no band annotated as B (+5) in the Fig. 3a of the revised manuscript. However, this annotation was present in the two previous versions of the manuscript. Apparently, the reviewer was looking at an older version of the manuscript, most likely at the very first version, where the marker lanes in Fig. 1 were not labeled yet.

Note: While we worked on the reviewer’s comments, we have noticed a typo in Fig. 3 that was corrected; the updated file was submitted together with the Reply to reviewer’s comment.